# Biological Function Analysis of MicroRNAs and Proteins in the Cerebrospinal Fluid of Patients with Parkinson’s Disease

**DOI:** 10.3390/ijms252413260

**Published:** 2024-12-10

**Authors:** Ji Su Hwang, Seok Gi Kim, Nimisha Pradeep George, Minjun Kwon, Yong Eun Jang, Sang Seop Lee, Gwang Lee

**Affiliations:** 1Department of Molecular Science and Technology, Ajou University, Suwon 16499, Republic of Korea; js3004@ajou.ac.kr (J.S.H.); rlatjrrl9977@ajou.ac.kr (S.G.K.); nimishapgeorge@ajou.ac.kr (N.P.G.); kmj936@ajou.ac.kr (M.K.); jye120@ajou.ac.kr (Y.E.J.); 2Department of Physiology, Ajou University School of Medicine, Suwon 16499, Republic of Korea; 3Department of Pharmacology, Inje University College of Medicine, Busan 47392, Republic of Korea; leess@inje.ac.kr

**Keywords:** cerebrospinal fluid, integrated omics, microRNA, Parkinson’s disease, proteomes

## Abstract

Parkinson’s disease (PD) is a progressive neurodegenerative disorder characterized by alpha-synuclein aggregation into Lewy bodies in the neurons. Cerebrospinal fluid (CSF) is considered the most suited source for investigating PD pathogenesis and identifying biomarkers. While microRNA (miRNA) profiling can aid in the investigation of post-transcriptional regulation in neurodegenerative diseases, information on miRNAs in the CSF of patients with PD remains limited. This review combines miRNA analysis with proteomic profiling to explore the collective impact of CSF miRNAs on the neurodegenerative mechanisms in PD. We constructed separate networks for altered miRNAs and proteomes using a bioinformatics method. Altered miRNAs were poorly linked to biological functions owing to limited information; however, changes in protein expression were strongly associated with biological functions. Subsequently, the networks were integrated for further analysis. In silico prediction from the integrated network revealed relationships between miRNAs and proteins, highlighting increased reactive oxygen species generation, neuronal loss, and neurodegeneration and suppressed ATP synthesis, mitochondrial function, and neurotransmitter release in PD. The approach suggests the potential of miRNAs as biomarkers for critical mechanisms underlying PD. The combined strategy could enhance our understanding of the complex biochemical networks of miRNAs in PD and support the development of diagnostic and therapeutic strategies for precision medicine.

## 1. Introduction

Parkinson’s disease (PD) is a chronic and progressive neurodegenerative disorder ranking second after Alzheimer’s disease (AD) [1]. PD is marked by the formation of Lewy bodies (cytoplasmic inclusions containing alpha-synuclein (α-syn)) along with the degeneration of dopaminergic (DA) neurons in the substantia nigra (SN) [2,3,4]. Postmortem studies of PD brains have shown a reduction in DA neuronal viability in the SN by approximately ≥30%, leading to motor impairments, such as tremors, rigidity, and bradykinesia, as well as non-motor features, including reduced olfactory ability, cognitive impairment, and psychiatric symptoms [5]. However, such hallmark symptoms of PD often overlap with those of other neurodegenerative diseases [6], posing a challenge for its early diagnosis. Moreover, current diagnostic practices primarily rely on clinical assessments [7], underscoring the necessity for refined diagnostic criteria.

Given the dopamine deficiency following DA neuronal death, medications such as levodopa, dopamine agonists, and monoamine oxidase B inhibitors have been developed to supplement dopamine action and have shown beneficial effects in the early stages of the disease [8,9,10]. However, unwanted side effects, such as hallucinations and impulse control disorders, remain a concern for patients [11,12]. Additionally, multiple system atrophy (MSA) shares several common impairments with PD but differs from it in terms of disease progression and response to dopaminergic treatment, hindering the fast and accurate diagnosis of PD [13,14]. Therefore, elucidation of the physiological mechanisms and development of critical biomarkers for discriminating PD from other neurodegenerative disorders are urgently required.

Numerous studies on the pathology of PD have identified several contributing factors, including genetic mutations, protein misfolding, oxidative stress, and mitochondrial dysfunction [15,16,17,18]. However, no definitive biomarker has been identified yet. To this end, a substantial number of studies have focused on changes in biomolecules in the brain and/or body fluids, including saliva, urine, blood, and cerebrospinal fluid (CSF) [19,20,21,22].

Among the fluids, CSF is increasingly regarded as a valuable source for research on PD owing to its proximity to the central nervous system (CNS) [23]. Unlike peripheral fluids, CSF directly interfaces with the extracellular space of the brain, allowing for an unrestricted two-way exchange of molecules between these regions [7]. In contrast, brain-derived proteins are typically not detected in blood-derived samples [24]. Approximately 20% of the proteins in CSF are derived specifically from brain cells, while the remainder originate from peripheral blood filtration [25]. Despite the relatively low proportion of brain-derived components, CSF is preferred over other sources to accurately mirror pathophysiological conditions [7]. In addition, CSF from PD patients was found to include toxic factors for dopaminergic neurons [26], highlighting its usefulness for PD research. Consequently, alterations in biomolecules, including transcripts, microRNAs (miRNAs), proteins, and metabolites, in the CSF are actively being evaluated using both experimental and bioinformatics methods [27,28,29,30].

Among the biomolecules of interest, miRNAs have emerged as important regulators of gene expression and translation [31]. These small non-coding RNAs, approximately 22 nucleotides in length [32], were first identified in *Caenorhabditis elegans* and regulate gene expression by interacting with complementary sequences on target mRNA through antisense RNA–RNA interactions [33]. miRNAs are implicated in various biological processes from the development, differentiation, proliferation, cell death, and immune systems [34]. In the context of neurodegenerative diseases, dysregulated miRNAs influence key pathways, such as neuroinflammation [35,36,37], oxidative stress [38,39,40], α-syn aggregation [41,42,43], and mitochondrial dysfunction [44,45,46]. These disruptions are particularly critical in PD, highlighting their significant role in disease onset and progression [47,48,49,50,51]. Consequently, significant efforts have been made to analyze various miRNAs in the CSF of patients with PD to identify potential miRNA biomarkers associated with these pathways [52,53,54,55,56,57]. However, the physiological roles of altered CSF miRNAs in patients with PD remain poorly understood, indicating the need for further research to elucidate their functions and diagnostic potential.

In this review, we aimed to employ an integrative strategy that combines miRNA and proteomic analyses to overcome the limitations of miRNAs and their unclear biological roles. Datasets from both miRNAomic and proteomic studies were collected from the published literature and analyzed to predict the biological functions related to PD using the bioinformatics tool Ingenuity Pathway Analysis (IPA, http://www.ingenuity.com, accessed on 28 October 2024) [58]. This integrative approach enabled the simultaneous examination of miRNAs and proteins, offering a more detailed understanding of the molecular mechanisms and biological functions underlying PD. Moreover, this analysis could uncover potential therapeutic targets by identifying miRNAs that modulate neuroprotective or neurodegenerative processes, thereby providing valuable insights into biomarker discovery and therapeutic development for PD. Herein, we divide this review into four sections as follows: (1) the advantages of CSF in PD diagnosis, (2) CSF miRNA in PD, (3) CSF protein in PD, and (4) the integration of miRNA and protein of the CSF of patients with PD.

## 2. Advantages of CSF in PD Diagnosis

Several studies have aimed to identify feasible biomarkers for the early diagnosis of PD by analyzing biomolecules, such as transcripts, miRNAs, proteins, and metabolites, in the blood of patients [20,59,60,61,62,63,64] and animal models [29,65,66,67]. Although blood-derived signatures are relatively easy to obtain, and associations be-tween molecular changes in the blood and PD continue to emerge [68], these studies are generally deemed less effective than CSF-derived studies in capturing the intricate mechanisms underlying PD [69].

CSF is primarily secreted by the choroid plexus [70,71]; its close contact with the extracellular space of the brain provides it distinct advantages for reflecting the state of the CNS [72]. Critical proteins involved in the pathology of PD and related disorders, such as tau, neuron-specific enolase, deglycase-1, chitinase-3-like protein 1, and α-syn, have been detected in CSF, underscoring the necessity of CSF analysis [44,73]. Notably, as aging progresses, the turnover and exchange of CSF components decreases, leading to the accumulation of proteins and other molecules, which may serve as potential biomarkers for PD [70,74]. Leveraging CSF analysis to elucidate PD mechanisms, discover biomarkers, and aid in the early diagnosis of PD may significantly enhance our understanding of its pathophysiology and lead to the development of more precise diagnostic tools, ultimately improving the outcomes of patients with PD.

## 3. CSF miRNA in PD

Emerging evidence suggests that dysregulated miRNAs, such as miR-7 and miR-153, are closely related to PD pathogenesis [75,76], and their presence in various body fluids underscores the potential of miRNAs as biomarkers of PD [77]. Dos Santos et al. identified three promising CSF miRNAs, namely miR-10b-5p, miR-22-3p, and miR-151a-3p, which are involved in PD pathogenesis, using small RNA sequencing and biomarker panel identification via machine learning techniques [78]. These findings suggest the diagnostic applicability of CSF miRNAs with their notable stability, quantifiability, and cost-effectiveness in the detection of neurodegenerative diseases, such as PD [79,80], MSA [56,81], amyotrophic lateral sclerosis (ALS) [82], and AD [83,84].

Building on this, miRNAs associated with PD pathology show limited overlap and different expression patterns in various biofluids [49,85]. Specifically, the miRNAs let-7g-3p, miR-10a-5p, miR-409-3p, miR-324-3p, and miR-205-5p were found to be upregulated in the CSF but not in other samples, such as whole blood, plasma, serum, and peripheral blood mononuclear cells [49,85]. Moreover, miR-7-5p was found to be downregulated in CSF but upregulated in serum [49]. This feature underscores the importance of analyzing miRNAs specifically in the CSF, which is structurally adjacent to the brain and more directly representative of the brain’s environment, to identify disease-specific biomarkers rather than relying on blood or peripheral fluids.

Subsequently, we compiled a list of miRNAs from CSF that were significantly altered and validated using various analytical methods to investigate PD pathologies based on the published literature (Table 1**)**. A total of 77 miRNAs were collected; 54 were upregulated and 23 were downregulated in the CSF of patients with PD. These changes in miRNAs were analyzed, and the miRNAomic network was constructed using IPA to explore their associations with biological functions and diseases related to PD.

Of these 77 miRNAs, 16 miRNAs were identified as contributors to seven different biological functions, including reactive oxygen species (ROS) production, ATP synthesis, mitochondrial function, the release of neurotransmitters, the loss of neurons, neurodegeneration, and PD (Figure 1A). Since the symbols for each miRNA that IPA software automatically recognized are different from Table 1, the details of 16 miRNAs in the miRNAomic network are provided in Appendix A. Then, based on the observation of 12 upregulated miRNAs and 4 downregulated miRNAs, the activation or inhibition of biological functions was predicted (Figure 1B). Notably, “synthesis of ATP” and “function of mitochondria” were predicted to be inhibited owing to the upregulation of miR-7a-5p, miR-16-5p, and mir-15, which directly inhibit these functions. Additionally, “neurodegeneration” was predicted to be activated by the upregulation of miR-17-5p. These findings are supported by previous studies. Specifically, miR-7a-5p, which is associated with ATP synthesis, has been reported to suppress cell proliferation upon upregulation, whereas its downregulation reduced apoptosis in non-small-cell lung cancer [91]. Similarly, miR-16-5p was shown to enhance mitochondrial function when its expression level was reduced in bladder cancer [92]. Furthermore, overexpression of mir-15 decreased ATP levels in rat ventricular myocytes [93]. Finally, miR-17-5p overexpression impaired TGF-beta signaling, leading to neurodegeneration in SH-SY5Y cells [94]. Taken together, these altered levels in the CSF are indicative of biological processes underlying PD. However, unfortunately, “generation of reactive oxygen species,” “loss of neurons,” and “Parkinson’s disease” were not predicted, and “release of neurotransmitter” was not linked to any miRNA. Although a thorough analysis of CSF miRNAs in PD pathology is still insufficient owing to their limited information, altered levels of CSF miRNAs revealed a strong association with PD, underscoring their potential as diagnostic biomarkers and therapeutic targets.

## 4. CSF Protein in PD

Proteomics research enables both quantitative and qualitative analyses of neurodegenerative diseases, particularly focusing on proteins expressed in the human brain in cases with AD, PD, frontotemporal dementia, and ALS [95,96,97]. This approach is particularly advantageous for identifying pathological alterations in proteins. Indeed, α-syn, a key protein in PD pathology, has been extensively studied using proteomics techniques, such as mass spectrometry, gel electrophoresis, and chromatography, revealing its crucial role in PD [98]. Advanced mass spectrometry technology has greatly improved sensitivity for detecting the molecules while reducing the sample size needed for high-throughput analysis, thereby facilitating not only the analysis of simple proteins but also the identification of post-translational modifications [99] that play a crucial role in regulating protein function in neuroscience [100]. Hondius et al. utilized laser microdissection combined with liquid chromatography–tandem mass spectrometry to analyze postmortem brain tissue [101], shedding light on the understanding of protein pathways involved in PD mechanisms.

To address the shortcomings in miRNAomic analysis, we obtained proteomic profiles of the CSF of patients with PD and identified 92 altered proteins (Table 2). This set, which includes 16 more proteins than our previous study of 76 biologically significant proteins [102], enabled a more comprehensive mechanistic analysis and enhanced the investigation of biological functions related to PD [103,104,105,106,107,108,109,110]. Of these, 43 were upregulated and 49 were downregulated.

Based on the identified profiles of proteins, a proteomic network was constructed with the same biological functions and diseases predicted in the miRNA network (Figure 2A). From this proteomic dataset, 40 proteins were identified as contributors in the network. Of these, 17 proteins showed increased levels, while 23 exhibited decreased levels. With these changes, “generation of reactive oxygen species,” “loss of neurons,” “neurodegeneration,” and “Parkinson’s disease” were predicted to be activated, while “synthesis of ATP,” “function of mitochondria,” and “release of neurotransmitter” were inhibited (Figure 2B). In particular, CO3, IL1B, PARK7, NGF, TGFA, TGFB1, and TNFA affected more than two functions, highlighting their significant roles in the pathogenesis and progression of PD. This proteomic network analysis indicated that the protein changes reflect PD conditions and that the interconnected proteins underlie the relationship between functions and diseases.

Notably, “release of neurotransmitter” was connected and predicted to be suppressed only in proteomics, not in miRNAome. This finding indicates that proteomic analysis of CSF can compensate for the limitations of miRNAome, as it captures direct evidence of altered protein expressions. Therefore, we incorporated two different omics together for synergetic effect to understand PD mechanisms and identify key molecules.

## 5. Integration of miRNA and Protein from the CSF of Patients with PD

The foundational concept of integrated omics was proposed by Dr. Leroy Hood [120,121], who proposed a systems biology approach that combines different types of omics data, such as genomics, epigenomics, transcriptomics, metabolomics, proteomics, and phosphoproteomics, to provide a comprehensive understanding of complex multifactorial biological systems [122]. The trans-omics concept of dynamic networks was proposed by Yugi et al. [123]. These approaches enable the identification of molecular signatures and potential therapeutic targets, facilitating precise diagnostics and advancing research across various fields of molecular biology, including neuroinflammation [124], nanotoxicity [125,126,127,128], mechanobiology [129], stem cell therapy [130,131,132], and neurodegenerative disease [133,134,135].

In this study, the miRNAome and proteome were integrated, and their relationships were considered using the latest IPA program, thereby identifying critical relationships between miRNAs and proteins (Figure 3A). Notably, the upregulation of miR-16-5p, which contributes to the inhibition of mitochondrial function, was associated with the regulation of AP2B1, CSF1, CLUS, SCF, LAMP2, IL6, and VEGFA, potentializing it as a biomarker with a significant contribution to PD pathology. Furthermore, upregulation of miR-7a-5p revealed an inhibitory effect on TGFA, which was found to activate “loss of neurons” and “neurodegeneration.” Next, the trends in biological functions and diseases in the integrated network were similar to those observed in both the miRNAomic and proteomic networks (Figure 3B). However, inhibition of “synthesis of ATP” and “function of mitochondria” was predicted to be more severe than in miRNAomic and proteomic networks by connecting the additional relationships between miRNAs and proteins. Therefore, a more profound prediction in the integrated network could provide more accurate and comprehensive insights. Furthermore, this approach enabled the investigation of the contributions of molecular expression patterns to the pathogenesis of PD (Figure 3C,D). Adhering to specific criteria (categories associated with “neurological” and |activation z-score| > 1), 97 upregulated molecules were found to induce immune neurological disorders, including “autoimmune neurological disorder” and “experimental autoimmune encephalomyelitis” and activate the “damage of neurons” (Figure 3C). Moreover, 72 downregulated molecules were associated with the activation of 16 disease and biological functions, indicating neuronal degeneration, impaired movement, and non-motor symptoms (Figure 3D). Among them, the exact same functions, “neurodegeneration” and “loss of neurons,” were predicted to be activated, highlighting the importance of CSF analysis in PD studies. Through this integrative approach, strong correlations between CSF components and biological functions were identified, underscoring the need for further research on the alterations in CSF to unravel PD-related mechanisms and disease progression.

In this review, we collected data on the changes in miRNAs and proteins in the CSF of patients with PD to analyze the molecular mechanisms associated with PD. Although CSF directly reflects the changes in the CNS, the discomfort and potential complications of the lumbar puncture process to obtain CSF samples limit its application in routine monitoring or large-scale diagnostic testing [136]. In contrast, peripheral blood sampling has the advantage of being relatively noninvasive and easily accessible. Despite obvious differences in the changes in miRNAs in the CSF and blood of patients with PD, certain miRNAs have been observed to exhibit identical patterns of increase or decrease in both the CSF and blood. For instance, Zhuang et al. reported that miR-125b levels are significantly higher in both CSF and plasma samples of patients with PD [90]. Tong et al. analyzed exosomes derived from the blood and CSF of patients with PD, highlighting five common miRNAs upregulated in both of the samples (miR-151a-5p, miR-24, mir-485-5p, mir-331-5p, and mir-214) [87]. These findings suggest that important miRNAs identified in both samples with the same pattern can be easily tested using blood-based assays, enabling early diagnosis and disease progression monitoring of PD. Therefore, comprehensive studies with miRNA levels in both CSF and blood-derived samples with integrative multi-omics analysis, such as transcriptomics, proteomics, phosphoproteomics, and metabolomics, with key miRNAs may elucidate the critical mechanisms of PD.

While this review highlights the potential of integrating multi-omics for PD analysis, it also has limitations. Data collection primarily relied on individual studies that were not corroborated by other research, emphasizing the need for follow-up studies. For instance, discrepancies in findings related to molecules such as miR-127-3p [55,89], miR-136-3p [86,87], and miR-433 [55,86] have hindered efforts to clarify mechanisms underlying PD. These inconsistencies suggest that more robust and reproducible studies are essential to resolve conflicting results and improve our understanding of molecular targets in PD.

Moreover, integrated omics approaches present unique challenges. Owing to the limited number of samples and the vast number of quantified molecules, they can often lead to false positives and negatives. Additionally, the analysis of time-series data and the distinction between true biological signals and noise in both targeted and untargeted approaches present significant hurdles. Therefore, effective data handling using multiple-omics data would be essential to ensure accurate reflection and reliability of biological phenotypes. Key steps include rigorous data filtering to remove noise and irrelevant features and thorough data cleaning to correct errors and manage missing values. This preprocessing ensures that downstream analyses, including statistical modeling and machine learning applications, yield meaningful biological insights and reliable predictions. For clustering multi-omics data, machine learning algorithms are useful in integrated omics research [124,133].

The application of artificial intelligence (AI) in integrated omics is a promising approach to address the challenges. Because integrated omics data are huge data, AI enhances their capabilities by processing vast amounts of complex biological time differences in omics data, non-obvious relationships between omics regimes, and high-dimensional omics data [137,138]. Additionally, AI can classify various omics data, uncover hidden patterns, perform feature selection, generate predictive models with high accuracy, accelerate biomarker discovery, and aid in clinical decision-making. In addition, the use of bioinformatics tools, such as IPA programs, can bring breakthroughs in this area. Identifying an apparent relationship between the different datasets within the biological context and independent statistical analysis of each omics regime is a challenge for researchers. In this context, we successfully integrated both datasets using the latest IPA program to construct a single and integrative network based on the pathological features of PD. By integrating omics data with AI approaches, researchers can derive deeper insights into the biological mechanisms in the CSF of patients with PD, develop more effective diagnostic tools, and investigate the pathophysiological mechanism of PD, thus advancing the field of neurodegenerative research.

## 6. Conclusions

This review highlights the importance of analyzing miRNAs and proteins in the CSF of patients with PD and proposes a powerful approach for understanding PD mechanisms by combining miRNAomic and proteomic data from the CSF using a computational method. Our functional network analyses of miRNAs and proteomes offer comprehensive insights into the post-transcriptional and translational regulations driving disease pathology including ROS production, ATP reduction, mitochondrial dysfunction, and inhibited release of neurotransmitters, resulting in the loss of neurons and neurodegeneration. This approach highlights key pathways involved in PD progression and identifies potential biomarkers and therapeutic targets in the CSF. As the field progresses, integrating multi-omics data and AI will provide a more in-depth understanding of PD and support the development of diagnostic and therapeutic strategies.

## Figures and Tables

**Figure 1 ijms-25-13260-f001:**
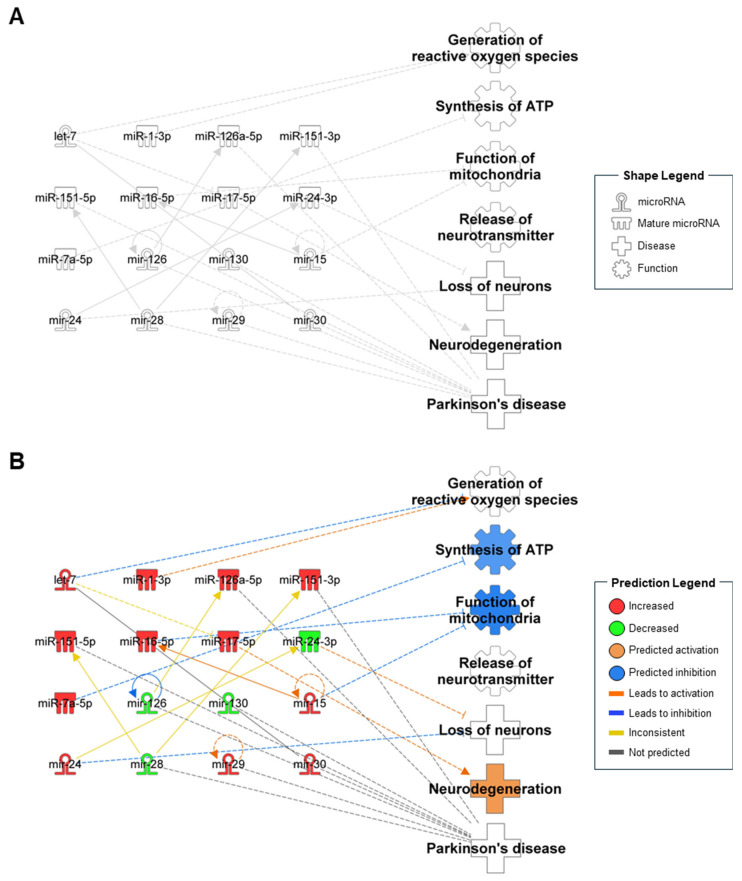
Biological function- and disease-related miRNAomic network from ingenuity pathway analysis. (**A**) Network of miRNAs (associated with pathological functions in Parkinson’s disease) from the cerebrospinal fluid (CSF) of patients. (**B**) In silico prediction of the miRNAomic network based on alterations of the miRNAs from CSF. The color intensity reflects the confidence of the prediction.

**Figure 2 ijms-25-13260-f002:**
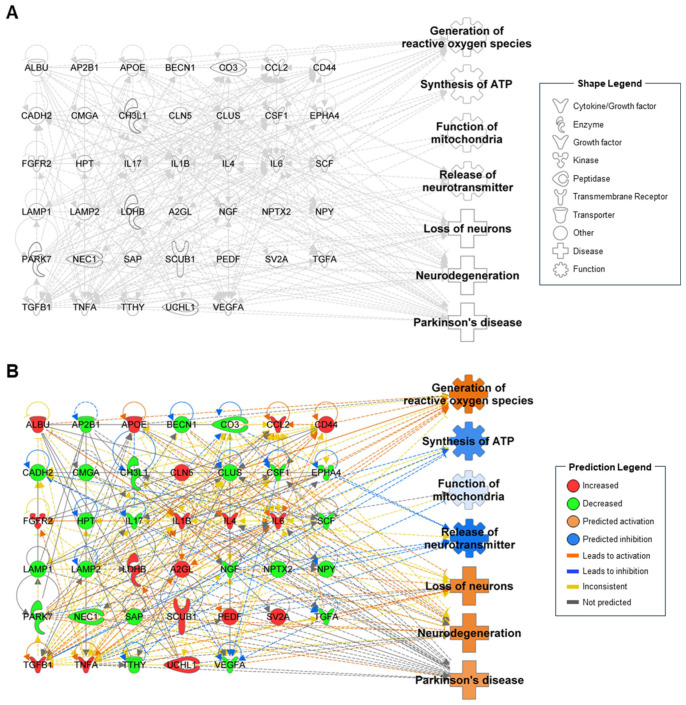
Biological function- and disease-related proteomic network from ingenuity pathway analysis. (**A**) Network of proteins associated with the pathological functions in Parkinson’s disease in the cerebrospinal fluid (CSF) of patients. (**B**) In silico prediction of the proteomic network based on the alterations of proteins from CSF. The color intensity reflects the confidence of the prediction.

**Figure 3 ijms-25-13260-f003:**
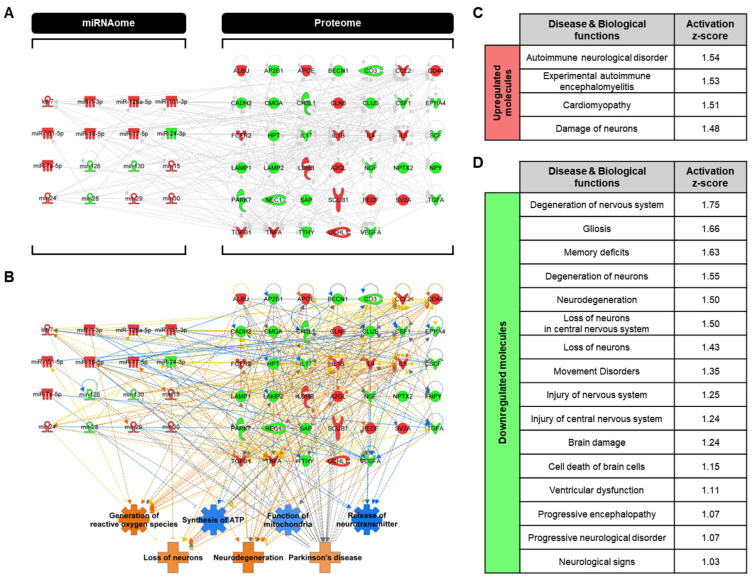
Integrated omics network analysis and prediction from ingenuity pathway analysis. (**A**) Integrated network of the miRNAome and proteome. (**B**) In silico prediction of integrated network based on alterations of miRNAs and proteins. (**C**,**D**) Analyses of diseases and biological functions in upregulated molecules (**C**) and downregulated molecules (**D**) from the integrated omics dataset. A positive activation z-score indicates the activation of the function. The color intensity reflects the confidence of the prediction.

**Table 1 ijms-25-13260-t001:** List of miRNAs that are altered in the cerebrospinal fluid of patients with Parkinson’s disease.

miRNA Symbol	miRBase ID	Expression Pattern	Analysis Method ^a^	Reference
miR-106b-5p	MIMAT0000680	Increased	RT-qPCR	[54]
miR-218-5p	MIMAT0000275	Increased	RT-qPCR	[54]
miR-7-5p	MIMAT0000252	Increased	RT-qPCR	[54]
miR-331-5p	MIMAT0004700	Increased	RT-qPCR, TaqMan low-density array	[54,86,87]
miR-34c-3p	MIMAT0004677	Increased	RT-qPCR	[54]
miR-30b-5p	MIMAT0000420	Increased	RT-qPCR	[54]
miR-30c-5p	MIMAT0000244	Increased	RT-qPCR	[54]
miR-19a-3p	MIMAT0000073	Increased	NGS	[55]
let-7g-3p	MIMAT0004584	Increased	NGS, TaqMan low-density array	[55,86]
mir-30b	MI0000441	Increased	RT-qPCR, TaqMan miRNA array	[86,87]
mir-16-2	MI0000115	Increased	RT-qPCR, TaqMan miRNA array	[86,87]
miR-205-5p	MIMAT0000266	Increased	RT-qPCR	[56]
miR-144-5p	MIMAT0004600	Increased	High-throughput deep sequencing, RT-qPCR	[88]
miR-200a-3p	MIMAT0000682	Increased	High-throughput deep sequencing, RT-qPCR	[87,88]
miR-542-3p	MIMAT0003389	Increased	High-throughput deep sequencing, RT-qPCR	[88]
miR-126-5p	MIMAT0000444	Increased	NGS	[89]
miR-126-3p	MIMAT0000445	Increased	NGS	[89]
miR-138-5p	MIMAT0000430	Increased	NGS	[89]
miR-9-5p	MIMAT0000441	Increased	NGS	[89]
miR-219a-2-3p	MIMAT0004675	Increased	NGS	[89]
miR-181a-5p	MIMAT0000256	Increased	NGS	[89]
miR-181b-5p	MIMAT0000257	Increased	NGS	[89]
miR-451a	MIMAT0001631	Increased	NGS	[52,89]
miR-486-5p	MIMAT0002177	Increased	NGS	[52,89]
miR-98-5p	MIMAT0000096	Increased	NGS	[89]
miR-144-3p	MIMAT0000436	Increased	NGS	[52,89]
miR-769-5p	MIMAT0003886	Increased	NGS	[89]
miR-211-5p	MIMAT0000268	Increased	NGS	[89]
miR-129-5p	MIMAT0000242	Increased	NGS	[89]
miR-16-5p	MIMAT0000069	Increased	NGS	[89]
miR-151a-5p	MIMAT0004697	Increased	RT-qPCR	[87]
miR-24	MI0000080, MI0000081	Increased	RT-qPCR	[87]
mir-214	MI0000290	Increased	RT-qPCR	[87]
let-7b	MI0000063	Increased	RT-qPCR	[87]
let-7f-1-3p	MIMAT0004486	Increased	RT-qPCR	[87]
miR-16	MI0000070	Increased	RT-qPCR	[87]
miR-200a-5p	MIMAT0001620	Increased	RT-qPCR	[87]
miR-26a-5p	MIMAT0000082	Increased	RT-qPCR	[87]
miR-29b-1	MI0000105	Increased	RT-qPCR	[87]
miR-122-5p	MIMAT0000421	Increased	NGS	[52]
miR-423-5p	MIMAT0004748	Increased	NGS	[52]
miR-151a-3p	MIMAT0000757	Increased	NGS	[52]
miR-320a	MI0000542	Increased	NGS	[52]
miR-320b	MIMAT0005792	Increased	NGS	[52]
miR-574-5p	MIMAT0004795	Increased	NGS	[52]
miR-206	MIMAT0000462	Increased	NGS	[52]
miR-1298-5p	MIMAT0005800	Increased	NGS	[52]
miR-1246	MIMAT0005898	Increased	NGS	[52]
miR-1307-3p	MIMAT0005951	Increased	NGS	[52]
miR-128-3p	MIMAT0000424	Increased	NGS	[52]
let-7a-5p	MIMAT0000062	Increased	NGS	[52]
let-7d-3p	MIMAT0004484	Increased	NGS	[52]
miR-4508	MIMAT0019045	Increased	NGS	[52]
miR-155-5p	MIMAT0000646	Increased	NGS	[52]
miR-99a-5p	MIMAT0000097	Decreased	RT-qPCR, NGS	[54,89]
miR-99b-5p	MIMAT0000689	Decreased	NGS	[89]
miR-100-5p	MIMAT0000098	Decreased	RT-qPCR	[54]
miR-145-5p	MIMAT0000437	Decreased	RT-qPCR	[54]
miR-92a-3p	MIMAT0000092	Decreased	RT-qPCR	[54]
miR-106a-5p	MIMAT0000103	Decreased	RT-qPCR	[54]
miR-128	MI0000727	Decreased	NGS	[55]
miR-431-3p	MIMAT0004757	Decreased	NGS	[55]
miR-212-3p	MIMAT0000269	Decreased	NGS	[55]
miR-1224-5p	MIMAT0005458	Decreased	NGS	[55]
miR-4448	MIMAT0018967	Decreased	NGS	[55]
mir-1	MI0000651	Decreased	TaqMan miRNA array, RT-qPCR	[86,87]
mir-126	MI0000471	Decreased	TaqMan miRNA array	[86]
mir-28	MI0000086	Decreased	TaqMan miRNA array	[86]
mir-301a	MI0000745	Decreased	TaqMan miRNA array	[86]
mir-29c	MI0000735	Decreased	TaqMan miRNA array	[86]
miR-24-3p	MIMAT0000080	Decreased	RT-qPCR	[56]
miR-626	MIMAT0003295	Decreased	RT-qPCR	[53]
miR-501-3p	MIMAT0004774	Decreased	NGS	[89]
miR-186-5p	MIMAT0000456	Decreased	NGS	[89]
miR-331-3p	MIMAT0000760	Decreased	RT-qPCR	[87]
miR-485-3p	MIMAT0002176	Decreased	RT-qPCR	[87]
miR-125b	MI0000446, MI0000470	Decreased	RT-qPCR	[90]

^a^ Abbreviations: RT-qPCR, reverse transcription-quantitative polymerase chain reaction; NGS, next-generation sequencing.

**Table 2 ijms-25-13260-t002:** List of proteins altered in the cerebrospinal fluid of patients with Parkinson’s disease.

Protein Symbol ^a^	Uniprot ID	Expression Pattern	Analysis Method ^b^	Reference
APOE	P02649	Increased	LC-MS/MS	[111]
ENPP2	Q13822	Increased	LC-MS/MS	[111]
CNDP1	Q96KN2	Increased	LC-MS/MS	[111]
LDHB	P07195	Increased	LC-MS/MS	[111]
PEDF	P36955	Increased	LC-MS/MS	[111]
ALBU	P02768	Increased	LC-MS/MS	[111]
APLD1	Q96LR9	Increased	LC-MS/MS	[89]
DNS2A	O00115	Increased	LC-MS/MS	[89]
KV401	P06312	Increased	LC-MS/MS	[89]
IPSP	P05154	Increased	LC-MS/MS	[89]
APOA4	P06727	Increased	LC-MS/MS	[89]
KAIN	P29622	Increased	LC-MS/MS	[89]
HV102	P23083	Increased	LC-MS/MS	[89]
IGHG4	P01861	Increased	LC-MS/MS	[89]
LV147	P01700	Increased	LC-MS/MS	[89]
KLKB1	P03952	Increased	LC-MS/MS	[89]
TM198	Q66K66	Increased	LC-MS/MS	[89]
CBPB2	Q96IY4	Increased	LC-MS/MS	[89]
RET4	P02753	Increased	LC-MS/MS	[89]
LSAMP	Q13449	Increased	MRM-LC-MS/MS	[112]
APOH	P02749	Increased	MRM-LC-MS/MS	[112]
C1QC	P02747	Increased	LC-MS/MS	[106]
SIAL	P21815	Increased	Protein microarray	[113]
CCL2	P13500	Increased	Protein microarray, Meta-analysis, Bead-based cytokine array	[110,113,114]
SV2A	Q7L0J3	Increased	Protein microarray	[113]
CCL14	Q16627	Increased	Protein microarray, LC-MS/MS	[115]
SCUBE1	Q8IWY4	Increased	Protein microarray, LC-MS/MS	[115]
OMD	Q99983	Increased	Protein microarray, LC-MS/MS	[115,116]
CLN5	O75503	Increased	Protein microarray, LC-MS/MS	[115]
MA1C1	Q9NR34	Increased	Protein microarray, LC-MS/MS	[115]
CD44	P16070	Increased	LC-MS/MS	[116]
IL6	P05231	Increased	Meta-analysis	[114]
TNFA	P01375	Increased	Meta-analysis	[114]
IL1B	P01584	Increased	Meta-analysis	[114]
CRP	P02741	Increased	Meta-analysis	[114]
CCL28	Q9NRJ3	Increased	Systematic review	[114]
IL4	P05112	Increased	Meta-analysis (ELISA)	[114]
TGFB1	P01137	Increased	Meta-analysis (ELISA)	[114]
A2GL	P02750	Increased	LC-IMS-MS, LC-MRM	[103]
POTEE	Q6S8J3	Increased	LC-IMS-MS	[103]
FGFR2	P21802	Increased	LC-IMS-MS	[103]
ANT3	P01008	Increased	LC-MS/MS	[106]
UCHL1	P09936	Increased	Human Neurology 4-Plex A Advantage Kit	[107]
SCG2	P13521	Decreased	LC-MS/MS	[89]
CLUS	P10909	Decreased	LC-MS/MS	[111]
CO3	P01024	Decreased	LC-MS/MS	[111]
CO4A	P0C0L4	Decreased	LC-MS/MS	[111]
CO4B	P0C0L5	Decreased	LC-MS/MS	[111]
DCD	P81605	Decreased	LC-MS/MS	[111]
HPT	P00738	Decreased	LC-MS/MS	[111]
TTHY	P02766	Decreased	LC-MS/MS	[111]
GOGA3	Q08378	Decreased	LC-MS/MS	[117]
APOB	P04114	Decreased	LC-MS/MS	[117]
VGF	O15240	Decreased	LC-MS/MS, HPLC-MS/MS	[89,106]
CADM2	Q8N3J6	Decreased	LC-MS/MS	[89]
PTPR2	Q92932	Decreased	LC-MS/MS, HPLC-MS/MS	[89,106]
NEC1	P29120	Decreased	LC-MS/MS	[89]
CMGA	P10645	Decreased	LC-MS/MS	[89]
VTM2A	Q8TAG5	Decreased	LC-MS/MS, HPLC-MS/MS	[89,106]
SLIK1	Q96PX8	Decreased	LC-MS/MS	[89]
NPY	P01303	Decreased	LC-MS/MS	[89]
7B2	P05408	Decreased	LC-MS/MS	[89]
APOC2	P02655	Decreased	LC-MS/MS	[89]
AP2B1	P63010	Decreased	LC-PRM-MS	[118]
CATF	Q9UBX1	Decreased	LC-PRM-MS	[118]
SAP3	P17900	Decreased	LC-PRM-MS	[118]
CGRE1	Q99674	Decreased	HPLC-MS/MS	[106]
TGON2	O43493	Decreased	HPLC-MS/MS	[106]
PARK7	Q99497	Decreased	Meta-analysis	[119]
CH3L1	P36222	Decreased	Meta-analysis	[114,119]
IL16	Q14005	Decreased	Systematic review	[114]
IL17	Q16552	Decreased	Systematic review	[114]
CCL8	P80075	Decreased	Systematic review	[114]
CCL23	P55773	Decreased	Systematic review	[114]
GROA	P09341	Decreased	Systematic review	[114]
NGF	P01138	Decreased	Systematic review	[114]
FGF19	O95750	Decreased	Systematic review	[114]
SCF	P21583	Decreased	Systematic review	[114]
CSF1	P09603	Decreased	Systematic review	[114]
PD1L1	Q9NZQ7	Decreased	Systematic review	[114]
VEGFA	P15692	Decreased	Systematic review	[114]
TGFA	P01135	Decreased	Meta-analysis (multiplex cytokine)	[114]
SAP	P07602	Decreased	LC-IMS-MS	[103]
NPTX2	P47972	Decreased	LC-MS/MS	[104]
NFL	P07196	Decreased	SIMOA	[105]
CADH2	P19022	Decreased	LC-MS/MS	[106]
EPHA4	P54764	Decreased	LC-MS/MS	[106]
QSOX1	O00391	Decreased	LC-MS/MS	[106]
MP3B2	A6NCE7	Decreased	ELISA	[108]
BECN1	Q14457	Decreased	ELISA	[108]
LAMP2	P13473	Decreased	ELISA	[108]
LAMP1	P11279	Decreased	Immunoblotting	[109]

^a^ The protein symbol indicates the human protein designated by UniProtKB. ^b^ Abbreviations: LC-MS/MS, liquid chromatography–tandem mass spectrometry; MRM-LC-MS/MS, multiple reaction monitoring–liquid chromatography–tandem mass spectrometry; ELISA, enzyme-linked immunosorbent assay; LC-IMS-MS, liquid chromatography–ion mobility spectrometry–mass spectrometry; LC-MRM, liquid chromatography–multiple reaction monitoring; LC-PRM-MS, liquid chromatography–parallel reaction monitoring–mass spectrometry; SIMOA, single-molecule array.

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
