# Peer review of "Biological Function Analysis of MicroRNAs and Proteins in the Cerebrospinal Fluid of Patients with Parkinson’s Disease"

_ijms, 2024, doi:10.3390/ijms252413260_

Round 1
Reviewer 1 Report
Comments and Suggestions for Authors
In this study, Ji et al. propose a review study on Parkinson's disease, mainly exploring the role of microRNAs and proteomes in cerebrospinal fluid in PD, and analyzing their collective impact on the pathogenesis of the disease. The article aims to reveal the impact of miRNAs in CSF on the neurodegenerative mechanism of PD by combining miRNA analysis and proteomic analysis, and constructs a network of altered miRNAs and proteomes to explore their relationship with PD related biological functions and diseases. However, before consider publishing in our journal, several revisions are required to improve the quality of manuscript.
Major Comments:
1. Line 66, The font size format of the letter “s” of “miRNAs” on line 66 is incorrect.
2. Line 145, “From this pool”, It is unclear whether “this” in refers to Figure 1 (A) or Figure 1 (B)? I think it would be more reasonable to place them after line 149.
3. Line 184, it is better to move the sentence of “based on the changes in proteins” to in Line 204 after Table 2?
4. Table 2 is too long and takes up about two pages, the author may compress it and put the original table into the Supplementary materials. In addition, why the font size of the last row in Table 2 is different from the other rows.
5. In lines 190-191, it is indicated that “synthesis of ATP”, “function of mitochondria” and “release of neurotransmitter” were inhibited (Figure 2B). In Figure 2 (B), blue indicates inhibition, but the color intensity of these three is not consistent. Does the color intensity have a special meaning, such as whether it represents the degree of inhibition? If so, should an explanation be made.
6. In Introduction section, the authors described “Parkinson’s disease (PD) is a progressive neurodegenerative disorder...”, it is recommended to add a paragraph to describe the unique advantages of cerebrospinal fluid (CSF) compared to other bodily fluids, thus strengthening the rationale for its selection.
7. In section “3. CSF miRNA in PD”, various PD-related miRNAs are mentioned, but the specific criteria and rationale for selecting these miRNAs are not provided. It would be beneficial to elaborate on the exact standards used to select these miRNAs and to discuss in depth how these miRNAs are involved in the biological processes of PD.
Author Response
Thank you for your sharp comments and valuable feedback. We carefully checked all your comments and revised the manuscript accordingly. The responses were provided right below each comment, with your comment in black, our responses in blue, and the changes made in the manuscript were colored in red.
Comments for Suggestions for Authors:
In this study, Ji et al. propose a review study on Parkinson's disease, mainly exploring the role of microRNAs and proteomes in cerebrospinal fluid in PD and analyzing their collective impact on the pathogenesis of the disease. The article aims to reveal the impact of miRNAs in CSF on the neurodegenerative mechanism of PD by combining miRNA analysis and proteomic analysis, and constructs a network of altered miRNAs and proteomes to explore their relationship with PD related biological functions and diseases. However, before consider publishing in our journal, several revisions are required to improve the quality of manuscript.
1. Line 66, The font size format of the letter “s” of “miRNAs” on line 66 is incorrect.
à According to your comment, we revised it to an identical format as well as the others.
Line 70, page 2
Among the biomolecules of interest, miRNAs have emerged as important regulators of gene expression and translation [31].
2. Line 145, “From this pool”, It is unclear whether “this” in refers to Figure 1 (A) or Figure 1 (B)? I think it would be more reasonable to place them after line 149.
à In line 145, “From this pool” refers to a dataset of 77 miRNAs used for analyzing the miRNAomic network. Our miRNAomic network revealed that 16 miRNAs were associated with seven biological functions which were representative features of Parkinson’s disease. However, we noticed that the following explanation of altered levels in these miRNAs might confuse readers about which figure was being referred to, so we deleted the mention of “Figure 1A” in line 139 and revised line 146 as below.
Additionally, we revised the names of miRNAs in Table 1, following the names provided from references, and the official ID identified by Ingenuity Pathway Analysis (IPA) software for 16 miRNAs from the miRNAomic network were provided in Supplementary Table 1.
Line 139-141, page 3
These changes in miRNAs were analyzed, and the miRNAomic network was constructed using IPA to explore their associations with biological functions and diseases related to PD.
Line 146-153, page 5
Of these 77 miRNAs, 16 miRNAs were identified as contributors to seven different biological functions, including the reactive oxygen species (ROS) production, ATP synthesis, mitochondrial function, release of neurotransmitter, loss of neurons, neurodegeneration, and PD (Figure 1A). Since the symbols for each miRNA that IPA software automatically recognized are different from Table 1, the details of 16 miRNAs in the miRNAomic network are provided in Supplementary Table 1. Then, based on the observation of 12 up-regulated miRNAs and 4 downregulated miRNAs, the activation or inhibition of biological functions was predicted (Figure 1B).
Table 1. List of miRNAs that are altered in the cerebrospinal fluid of patients with Parkinson’s disease.
|
miRNA symbol |
miRBase ID |
Expression pattern |
Analysis methoda |
Reference |
|
miR-106b-5p |
MIMAT0000680 |
Increased |
RT-qPCR, TaqMan low-density array |
[54] |
|
miR-218-5p |
MIMAT0000275 |
Increased |
RT-qPCR, TaqMan low-density array |
[54] |
|
miR-7-5p |
MIMAT0000252 |
Increased |
RT-qPCR, TaqMan low-density array |
[54] |
|
miR-331-5p |
MIMAT0004700 |
Increased |
RT-qPCR, TaqMan low-density array |
[54,86,87] |
|
miR-34c-3p |
MIMAT0004677 |
Increased |
RT-qPCR, TaqMan low-density array |
[54] |
|
miR-30b-5p |
MIMAT0000420 |
Increased |
RT-qPCR, TaqMan low-density array |
[54] |
|
miR-30c-5p |
MIMAT0000244 |
Increased |
RT-qPCR, TaqMan low-density array |
[54] |
|
miR-19a-3p |
MIMAT0000073 |
Increased |
RT-qPCR, TaqMan low-density array |
[55] |
|
let-7g-3p |
MIMAT0004584 |
Increased |
RT-qPCR, TaqMan low-density array |
[55,86] |
|
mir-30b |
MI0000441 |
Increased |
RT-qPCR, TaqMan miRNA array |
[86,87] |
|
mir-16-2 |
MI0000115 |
Increased |
RT-qPCR, TaqMan miRNA array |
[86,87] |
|
miR-205-5p |
MIMAT0000266 |
Increased |
RT-qPCR |
[56] |
|
miR-144-5p |
MIMAT0004600 |
Increased |
High-throughput deep sequencing, RT-qPCR |
[88] |
|
miR-200a-3p |
MIMAT0000682 |
Increased |
High-throughput deep sequencing, RT-qPCR |
[87,88] |
|
miR-542-3p |
MIMAT0003389 |
Increased |
High-throughput deep sequencing, RT-qPCR |
[88] |
|
miR-126-5p |
MIMAT0000444 |
Increased |
NGS |
[89] |
|
miR-126-3p |
MIMAT0000445 |
Increased |
NGS |
[89] |
|
miR-138-5p |
MIMAT0000430 |
Increased |
NGS |
[89] |
|
miR-9-5p |
MIMAT0000441 |
Increased |
NGS |
[89] |
|
miR-219a-2-3p |
MIMAT0004675 |
Increased |
NGS |
[89] |
|
miR-181a-5p |
MIMAT0000256 |
Increased |
NGS |
[89] |
|
miR-181b-5p |
MIMAT0000257 |
Increased |
NGS |
[89] |
|
miR-451a |
MIMAT0001631 |
Increased |
NGS |
[52,89] |
|
miR-486-5p |
MIMAT0002177 |
Increased |
NGS |
[52,89] |
|
miR-98-5p |
MIMAT0000096 |
Increased |
NGS |
[89] |
|
miR-144-3p |
MIMAT0000436 |
Increased |
NGS |
[52,89] |
|
miR-769-5p |
MIMAT0003886 |
Increased |
NGS |
[89] |
|
miR-211-5p |
MIMAT0000268 |
Increased |
NGS |
[89] |
|
miR-129-5p |
MIMAT0000242 |
Increased |
NGS |
[89] |
|
miR-16-5p |
MIMAT0000069 |
Increased |
NGS |
[89] |
|
miR-151a-5p |
MIMAT0004697 |
Increased |
RT-qPCR |
[87] |
|
miR-24 |
MI0000080, MI0000081 |
Increased |
RT-qPCR |
[87] |
|
mir-214 |
MI0000290 |
Increased |
RT-qPCR |
[87] |
|
let-7b |
MI0000063 |
Increased |
RT-qPCR |
[87] |
|
let-7f-1-3p |
MIMAT0004486 |
Increased |
RT-qPCR |
[87] |
|
miR-16 |
MI0000070 |
Increased |
RT-qPCR |
[87] |
|
miR-200a-5p |
MIMAT0001620 |
Increased |
RT-qPCR |
[87] |
|
miR-26a-5p |
MIMAT0000082 |
Increased |
RT-qPCR |
[87] |
|
miR-29b-1 |
MI0000105 |
Increased |
RT-qPCR |
[87] |
|
miR-122-5p |
MIMAT0000421 |
Increased |
NGS |
[52] |
|
miR-423-5p |
MIMAT0004748 |
Increased |
NGS |
[52] |
|
miR-151a-3p |
MIMAT0000757 |
Increased |
NGS |
[52] |
|
miR-320a |
MI0000542 |
Increased |
NGS |
[52] |
|
miR-320b |
MIMAT0005792 |
Increased |
NGS |
[52] |
|
miR-574-5p |
MIMAT0004795 |
Increased |
NGS |
[52] |
|
miR-206 |
MIMAT0000462 |
Increased |
NGS |
[52] |
|
miR-1298-5p |
MIMAT0005800 |
Increased |
NGS |
[52] |
|
miR-1246 |
MIMAT0005898 |
Increased |
NGS |
[52] |
|
miR-1307-3p |
MIMAT0005951 |
Increased |
NGS |
[52] |
|
miR-128-3p |
MIMAT0000424 |
Increased |
NGS |
[52] |
|
let-7a-5p |
MIMAT0000062 |
Increased |
NGS |
[52] |
|
let-7d-3p |
MIMAT0004484 |
Increased |
NGS |
[52] |
|
miR-4508 |
MIMAT0019045 |
Increased |
NGS |
[52] |
|
miR-155-5p |
MIMAT0000646 |
Increased |
NGS |
[52] |
|
miR-99a-5p |
MIMAT0000097 |
Decreased |
RT-qPCR, TaqMan low-density array, NGS |
[54,89] |
|
miR-99b-5p |
MIMAT0000689 |
Decreased |
RT-qPCR, TaqMan low-density array |
[54] |
|
miR-100-5p |
MIMAT0000098 |
Decreased |
RT-qPCR, TaqMan low-density array |
[54] |
|
miR-145-5p |
MIMAT0000437 |
Decreased |
RT-qPCR, TaqMan low-density array |
[54] |
|
miR-92a-3p |
MIMAT0000092 |
Decreased |
RT-qPCR, TaqMan low-density array |
[54] |
|
miR-106a-5p |
MIMAT0000103 |
Decreased |
RT-qPCR, TaqMan low-density array |
[54] |
|
miR-128 |
MI0000727 |
Decreased |
TaqMan low-density array |
[55] |
|
miR-431-3p |
MIMAT0004757 |
Decreased |
TaqMan low-density array |
[55] |
|
miR-212-3p |
MIMAT0000269 |
Decreased |
TaqMan low-density array |
[55] |
|
miR-1224-5p |
MIMAT0005458 |
Decreased |
TaqMan low-density array |
[55] |
|
miR-4448 |
MIMAT0018967 |
Decreased |
TaqMan low-density array |
[55] |
|
mir-1 |
MI0000651 |
Decreased |
TaqMan miRNA array, RT-qPCR |
[86,87] |
|
mir-126 |
MI0000471 |
Decreased |
TaqMan miRNA array |
[86] |
|
mir-28 |
MI0000086 |
Decreased |
TaqMan miRNA array |
[86] |
|
mir-301a |
MI0000745 |
Decreased |
TaqMan miRNA array |
[86] |
|
mir-29c |
MI0000735 |
Decreased |
TaqMan miRNA array |
[86] |
|
miR-24-3p |
MIMAT0000080 |
Decreased |
RT-qPCR |
[56] |
|
miR-626 |
MIMAT0003295 |
Decreased |
RT-qPCR |
[53] |
|
miR-501-3p |
MIMAT0004774 |
Decreased |
RT-qPCR |
[89] |
|
miR-186-5p |
MIMAT0000456 |
Decreased |
RT-qPCR |
[89] |
|
miR-331-3p |
MIMAT0000760 |
Decreased |
RT-qPCR |
[89] |
|
miR-485-3p |
MIMAT0002176 |
Decreased |
RT-qPCR |
[89] |
|
miR-125b |
MI0000446, MI0000470 |
Decreased |
RT-qPCR |
[90] |
aAbbreviations: RT-qPCR, reverse transcription-quantitative polymerase chain reaction; NGS, next-generation sequencing.
Supplementary Table 1. List of miRNA symbols identified by Ingenuity Pathway Analysis (IPA) software for 16 miRNAs in miRNAomic network.
|
No. |
IPA symbol |
miRBase ID |
miRNA symbol |
Reference |
|
1 |
let-7 |
MI0000063 |
let-7b |
[87] |
|
2 |
miR-1-3p |
MIMAT0000462 |
miR-206 |
[52] |
|
3 |
mir-126 |
MI0000471 |
mir-126 |
[86] |
|
4 |
miR-126a-5p |
MIMAT0000444 |
miR-126-5p |
[89] |
|
5 |
mir-130 |
MI0000745 |
mir-301a |
[86] |
|
6 |
mir-15 |
MI0000070 |
miR-16 |
[87] |
|
7 |
miR-151-3p |
MIMAT0000757 |
miR-151a-3p |
[52] |
|
8 |
miR-151-5p |
MIMAT0004697 |
miR-151a-5p |
[87] |
|
9 |
miR-16-5p |
MIMAT0000069 |
miR-16-5p |
[89] |
|
10 |
miR-17-5p |
MIMAT0000680 |
miR-106b-5p |
[54] |
|
11 |
mir-24 |
MI0000080 |
miR-24 |
[87] |
|
12 |
miR-24-3p |
MIMAT0000080 |
miR-24-3p |
[56] |
|
13 |
mir-28 |
MI0000086 |
mir-28 |
[86] |
|
14 |
mir-29 |
MI0000105 |
miR-29b-1 |
[87] |
|
15 |
mir-30 |
MI0000441 |
mir-30b |
[86,87] |
|
16 |
miR-7a-5p |
MIMAT0000252 |
miR-7-5p |
[54] |
3. Line 184, it is better to move the sentence of “based on the changes in proteins” to in Line 204 after Table 2?
à We agree with your opinion, therefore we revised and moved the sentence in line 184 after Table 2 to line 205. Furthermore, the explanation following was edited for improved readability as below.
Line 205-212, page 9
Based on the identified profiles of proteins, a proteomic network was constructed with the same biological functions and diseases predicted in the miRNA network (Figure 2A). From this proteomic dataset, 40 proteins were identified as contributors in the network. Of these, 17 proteins showed increased levels, while 23 exhibited decreased levels. With these changes, “generation of reactive oxygen species,” “loss of neurons,” “neurodegeneration,” and “Parkinson’s disease” were predicted to be activated, while “synthesis of ATP,” “function of mitochondria,” and “release of neurotransmitter” were inhibited (Figure 2B).
4. Table 2 is too long and takes up about two pages, the author may compress it and put the original table into the Supplementary materials. In addition, why the font size of the last row in Table 2 is different from the other rows.
à According to your suggestion, we unified the format of the last row in Table 2. However, we believe that this information is essential for maintaining the accessibility of the data, so the Table 2 was left in the main manuscript. To address the length issue, we reduced the font size to make it fit within two pages. Additionally, the font size of Table 1 was also adjusted to ensure a consistent format across the manuscript.
Table 1. List of miRNAs that are altered in the cerebrospinal fluid of patients with Parkinson’s disease.
|
miRNA symbol |
miRBase ID |
Expression pattern |
Analysis methoda |
Reference |
|
miR-106b-5p |
MIMAT0000680 |
Increased |
RT-qPCR, TaqMan low-density array |
[54] |
|
miR-218-5p |
MIMAT0000275 |
Increased |
RT-qPCR, TaqMan low-density array |
[54] |
|
miR-7-5p |
MIMAT0000252 |
Increased |
RT-qPCR, TaqMan low-density array |
[54] |
|
miR-331-5p |
MIMAT0004700 |
Increased |
RT-qPCR, TaqMan low-density array |
[54,86,87] |
|
miR-34c-3p |
MIMAT0004677 |
Increased |
RT-qPCR, TaqMan low-density array |
[54] |
|
miR-30b-5p |
MIMAT0000420 |
Increased |
RT-qPCR, TaqMan low-density array |
[54] |
|
miR-30c-5p |
MIMAT0000244 |
Increased |
RT-qPCR, TaqMan low-density array |
[54] |
|
miR-19a-3p |
MIMAT0000073 |
Increased |
RT-qPCR, TaqMan low-density array |
[55] |
|
let-7g-3p |
MIMAT0004584 |
Increased |
RT-qPCR, TaqMan low-density array |
[55,86] |
|
mir-30b |
MI0000441 |
Increased |
RT-qPCR, TaqMan miRNA array |
[86,87] |
|
mir-16-2 |
MI0000115 |
Increased |
RT-qPCR, TaqMan miRNA array |
[86,87] |
|
miR-205-5p |
MIMAT0000266 |
Increased |
RT-qPCR |
[56] |
|
miR-144-5p |
MIMAT0004600 |
Increased |
High-throughput deep sequencing, RT-qPCR |
[88] |
|
miR-200a-3p |
MIMAT0000682 |
Increased |
High-throughput deep sequencing, RT-qPCR |
[87,88] |
|
miR-542-3p |
MIMAT0003389 |
Increased |
High-throughput deep sequencing, RT-qPCR |
[88] |
|
miR-126-5p |
MIMAT0000444 |
Increased |
NGS |
[89] |
|
miR-126-3p |
MIMAT0000445 |
Increased |
NGS |
[89] |
|
miR-138-5p |
MIMAT0000430 |
Increased |
NGS |
[89] |
|
miR-9-5p |
MIMAT0000441 |
Increased |
NGS |
[89] |
|
miR-219a-2-3p |
MIMAT0004675 |
Increased |
NGS |
[89] |
|
miR-181a-5p |
MIMAT0000256 |
Increased |
NGS |
[89] |
|
miR-181b-5p |
MIMAT0000257 |
Increased |
NGS |
[89] |
|
miR-451a |
MIMAT0001631 |
Increased |
NGS |
[52,89] |
|
miR-486-5p |
MIMAT0002177 |
Increased |
NGS |
[52,89] |
|
miR-98-5p |
MIMAT0000096 |
Increased |
NGS |
[89] |
|
miR-144-3p |
MIMAT0000436 |
Increased |
NGS |
[52,89] |
|
miR-769-5p |
MIMAT0003886 |
Increased |
NGS |
[89] |
|
miR-211-5p |
MIMAT0000268 |
Increased |
NGS |
[89] |
|
miR-129-5p |
MIMAT0000242 |
Increased |
NGS |
[89] |
|
miR-16-5p |
MIMAT0000069 |
Increased |
NGS |
[89] |
|
miR-151a-5p |
MIMAT0004697 |
Increased |
RT-qPCR |
[87] |
|
miR-24 |
MI0000080, MI0000081 |
Increased |
RT-qPCR |
[87] |
|
mir-214 |
MI0000290 |
Increased |
RT-qPCR |
[87] |
|
let-7b |
MI0000063 |
Increased |
RT-qPCR |
[87] |
|
let-7f-1-3p |
MIMAT0004486 |
Increased |
RT-qPCR |
[87] |
|
miR-16 |
MI0000070 |
Increased |
RT-qPCR |
[87] |
|
miR-200a-5p |
MIMAT0001620 |
Increased |
RT-qPCR |
[87] |
|
miR-26a-5p |
MIMAT0000082 |
Increased |
RT-qPCR |
[87] |
|
miR-29b-1 |
MI0000105 |
Increased |
RT-qPCR |
[87] |
|
miR-122-5p |
MIMAT0000421 |
Increased |
NGS |
[52] |
|
miR-423-5p |
MIMAT0004748 |
Increased |
NGS |
[52] |
|
miR-151a-3p |
MIMAT0000757 |
Increased |
NGS |
[52] |
|
miR-320a |
MI0000542 |
Increased |
NGS |
[52] |
|
miR-320b |
MIMAT0005792 |
Increased |
NGS |
[52] |
|
miR-574-5p |
MIMAT0004795 |
Increased |
NGS |
[52] |
|
miR-206 |
MIMAT0000462 |
Increased |
NGS |
[52] |
|
miR-1298-5p |
MIMAT0005800 |
Increased |
NGS |
[52] |
|
miR-1246 |
MIMAT0005898 |
Increased |
NGS |
[52] |
|
miR-1307-3p |
MIMAT0005951 |
Increased |
NGS |
[52] |
|
miR-128-3p |
MIMAT0000424 |
Increased |
NGS |
[52] |
|
let-7a-5p |
MIMAT0000062 |
Increased |
NGS |
[52] |
|
let-7d-3p |
MIMAT0004484 |
Increased |
NGS |
[52] |
|
miR-4508 |
MIMAT0019045 |
Increased |
NGS |
[52] |
|
miR-155-5p |
MIMAT0000646 |
Increased |
NGS |
[52] |
|
miR-99a-5p |
MIMAT0000097 |
Decreased |
RT-qPCR, TaqMan low-density array, NGS |
[54,89] |
|
miR-99b-5p |
MIMAT0000689 |
Decreased |
RT-qPCR, TaqMan low-density array |
[54] |
|
miR-100-5p |
MIMAT0000098 |
Decreased |
RT-qPCR, TaqMan low-density array |
[54] |
|
miR-145-5p |
MIMAT0000437 |
Decreased |
RT-qPCR, TaqMan low-density array |
[54] |
|
miR-92a-3p |
MIMAT0000092 |
Decreased |
RT-qPCR, TaqMan low-density array |
[54] |
|
miR-106a-5p |
MIMAT0000103 |
Decreased |
RT-qPCR, TaqMan low-density array |
[54] |
|
miR-128 |
MI0000727 |
Decreased |
TaqMan low-density array |
[55] |
|
miR-431-3p |
MIMAT0004757 |
Decreased |
TaqMan low-density array |
[55] |
|
miR-212-3p |
MIMAT0000269 |
Decreased |
TaqMan low-density array |
[55] |
|
miR-1224-5p |
MIMAT0005458 |
Decreased |
TaqMan low-density array |
[55] |
|
miR-4448 |
MIMAT0018967 |
Decreased |
TaqMan low-density array |
[55] |
|
mir-1 |
MI0000651 |
Decreased |
TaqMan miRNA array, RT-qPCR |
[86,87] |
|
mir-126 |
MI0000471 |
Decreased |
TaqMan miRNA array |
[86] |
|
mir-28 |
MI0000086 |
Decreased |
TaqMan miRNA array |
[86] |
|
mir-301a |
MI0000745 |
Decreased |
TaqMan miRNA array |
[86] |
|
mir-29c |
MI0000735 |
Decreased |
TaqMan miRNA array |
[86] |
|
miR-24-3p |
MIMAT0000080 |
Decreased |
RT-qPCR |
[56] |
|
miR-626 |
MIMAT0003295 |
Decreased |
RT-qPCR |
[53] |
|
miR-501-3p |
MIMAT0004774 |
Decreased |
RT-qPCR |
[89] |
|
miR-186-5p |
MIMAT0000456 |
Decreased |
RT-qPCR |
[89] |
|
miR-331-3p |
MIMAT0000760 |
Decreased |
RT-qPCR |
[89] |
|
miR-485-3p |
MIMAT0002176 |
Decreased |
RT-qPCR |
[89] |
|
miR-125b |
MI0000446, MI0000470 |
Decreased |
RT-qPCR |
[90] |
aAbbreviations: RT-qPCR, reverse transcription-quantitative polymerase chain reaction; NGS, next-generation sequencing.
Table 2. List of proteins altered in the cerebrospinal fluid of patients with Parkinson’s disease.
|
Protein symbola |
Uniprot ID |
Expression pattern |
Analysis methodb |
Reference |
|
APOE |
P02649 |
Increased |
LC-MS/MS |
[111] |
|
ENPP2 |
Q13822 |
Increased |
LC-MS/MS |
[111] |
|
CNDP1 |
Q96KN2 |
Increased |
LC-MS/MS |
[111] |
|
LDHB |
P07195 |
Increased |
LC-MS/MS |
[111] |
|
PEDF |
P36955 |
Increased |
LC-MS/MS |
[111] |
|
ALBU |
P02768 |
Increased |
LC-MS/MS |
[111] |
|
APLD1 |
Q96LR9 |
Increased |
LC-MS/MS |
[89] |
|
DNS2A |
O00115 |
Increased |
LC-MS/MS |
[89] |
|
KV401 |
P06312 |
Increased |
LC-MS/MS |
[89] |
|
IPSP |
P05154 |
Increased |
LC-MS/MS |
[89] |
|
APOA4 |
P06727 |
Increased |
LC-MS/MS |
[89] |
|
KAIN |
P29622 |
Increased |
LC-MS/MS |
[89] |
|
HV102 |
P23083 |
Increased |
LC-MS/MS |
[89] |
|
IGHG4 |
P01861 |
Increased |
LC-MS/MS |
[89] |
|
LV147 |
P01700 |
Increased |
LC-MS/MS |
[89] |
|
KLKB1 |
P03952 |
Increased |
LC-MS/MS |
[89] |
|
TM198 |
Q66K66 |
Increased |
LC-MS/MS |
[89] |
|
CBPB2 |
Q96IY4 |
Increased |
LC-MS/MS |
[89] |
|
RET4 |
P02753 |
Increased |
LC-MS/MS |
[89] |
|
LSAMP |
Q13449 |
Increased |
MRM-LC-MS/MS |
[112] |
|
APOH |
P02749 |
Increased |
MRM-LC-MS/MS |
[112] |
|
C1QC |
P02747 |
Increased |
LC-MS/MS |
[106] |
|
SIAL |
P21815 |
Increased |
Protein microarray |
[113] |
|
CCL2 |
P13500 |
Increased |
Protein microarray, Meta-analysis, Bead-based cytokine array |
[110,113,114] |
|
SV2A |
Q7L0J3 |
Increased |
Protein microarray |
[113] |
|
CCL14 |
Q16627 |
Increased |
Protein microarray, LC-MS/MS |
[115] |
|
SCUBE1 |
Q8IWY4 |
Increased |
Protein microarray, LC-MS/MS |
[115,116] |
|
OMD |
Q99983 |
Increased |
Protein microarray, LC-MS/MS |
[115] |
|
CLN5 |
O75503 |
Increased |
Protein microarray, LC-MS/MS |
[115] |
|
MA1C1 |
Q9NR34 |
Increased |
Protein microarray, LC-MS/MS |
[116] |
|
CD44 |
P16070 |
Increased |
LC-MS/MS |
[116] |
|
IL6 |
P05231 |
Increased |
Meta-analysis |
[114] |
|
TNFA |
P01375 |
Increased |
Meta-analysis |
[114] |
|
IL1B |
P01584 |
Increased |
Meta-analysis |
[114] |
|
CRP |
P02741 |
Increased |
Meta-analysis |
[114] |
|
CCL28 |
Q9NRJ3 |
Increased |
Systematic review |
[114] |
|
IL4 |
P05112 |
Increased |
Meta-analysis (ELISA) |
[114] |
|
TGFB1 |
P01137 |
Increased |
Meta-analysis (ELISA) |
[114] |
|
A2GL |
P02750 |
Increased |
LC-IMS-MS, LC-MRM |
[103] |
|
POTEE |
Q6S8J3 |
Increased |
LC-IMS-MS |
[105] |
|
FGFR2 |
P21802 |
Increased |
LC-IMS-MS |
[105] |
|
ANT3 |
P01008 |
Increased |
LC-MS/MS |
[106] |
|
UCHL1 |
P09936 |
Increased |
Human Neurology 4-Plex A Advantage Kit |
[107] |
|
SCG2 |
P13521 |
Decreased |
HPLC-MS/MS, LC-MS/MS |
[56,89] |
|
CLUS |
P10909 |
Decreased |
LC-MS/MS |
[111] |
|
CO3 |
P01024 |
Decreased |
LC-MS/MS |
[111] |
|
CO4A |
P0C0L4 |
Decreased |
LC-MS/MS |
[111] |
|
CO4B |
P0C0L5 |
Decreased |
LC-MS/MS |
[111] |
|
DCD |
P81605 |
Decreased |
LC-MS/MS |
[111] |
|
HPT |
P00738 |
Decreased |
LC-MS/MS |
[111] |
|
TTHY |
P02766 |
Decreased |
LC-MS/MS |
[111] |
|
GOGA3 |
Q08378 |
Decreased |
LC-MS/MS |
[117] |
|
APOB |
P04114 |
Decreased |
LC-MS/MS |
[117] |
|
VGF |
O15240 |
Decreased |
LC-MS/MS, HPLC-MS/MS |
[89,106] |
|
CADM2 |
Q8N3J6 |
Decreased |
LC-MS/MS |
[89] |
|
PTPR2 |
Q92932 |
Decreased |
LC-MS/MS, HPLC-MS/MS |
[89,106] |
|
NEC1 |
P29120 |
Decreased |
LC-MS/MS |
[89] |
|
CMGA |
P10645 |
Decreased |
LC-MS/MS |
[89] |
|
VTM2A |
Q8TAG5 |
Decreased |
LC-MS/MS, HPLC-MS/MS |
[89,106] |
|
SLIK1 |
Q96PX8 |
Decreased |
LC-MS/MS |
[89] |
|
NPY |
P01303 |
Decreased |
LC-MS/MS |
[89] |
|
7B2 |
P05408 |
Decreased |
LC-MS/MS |
[89] |
|
APOC2 |
P02655 |
Decreased |
LC-MS/MS |
[89] |
|
AP2B1 |
P63010 |
Decreased |
LC-PRM-MS |
[118] |
|
CATF |
Q9UBX1 |
Decreased |
LC-PRM-MS |
[118] |
|
SAP3 |
P17900 |
Decreased |
LC-PRM-MS |
[118] |
|
CGRE1 |
Q99674 |
Decreased |
HPLC-MS/MS |
[106] |
|
TGON2 |
O43493 |
Decreased |
HPLC-MS/MS |
[106] |
|
PARK7 |
Q99497 |
Decreased |
Meta-analysis |
[119] |
|
CH3L1 |
P36222 |
Decreased |
Meta-analysis |
[114,119] |
|
IL16 |
Q14005 |
Decreased |
Systematic review |
[114] |
|
IL17 |
Q16552 |
Decreased |
Systematic review |
[114] |
|
CCL8 |
P80075 |
Decreased |
Systematic review |
[114] |
|
CCL23 |
P55773 |
Decreased |
Systematic review |
[114] |
|
GROA |
P09341 |
Decreased |
Systematic review |
[114] |
|
NGF |
P01138 |
Decreased |
Systematic review |
[114] |
|
FGF19 |
O95750 |
Decreased |
Systematic review |
[114] |
|
SCF |
P21583 |
Decreased |
Systematic review |
[114] |
|
CSF1 |
P09603 |
Decreased |
Systematic review |
[114] |
|
PD1L1 |
Q9NZQ7 |
Decreased |
Systematic review |
[114] |
|
VEGFA |
P15692 |
Decreased |
Systematic review |
[114] |
|
TGFA |
P01135 |
Decreased |
Meta-analysis (multiplex cytokine) |
[114] |
|
SAP |
P07602 |
Decreased |
LC-IMS-MS |
[103] |
|
NPTX2 |
P47972 |
Decreased |
LC-MS/MS |
[104] |
|
NFL |
P07196 |
Decreased |
SIMOA |
[105] |
|
CADH2 |
P19022 |
Decreased |
LC-MS/MS |
[106] |
|
EPHA4 |
P54764 |
Decreased |
LC-MS/MS |
[106] |
|
QSOX1 |
O00391 |
Decreased |
LC-MS/MS |
[106] |
|
MP3B2 |
A6NCE7 |
Decreased |
ELISA |
[108] |
|
BECN1 |
Q14457 |
Decreased |
ELISA |
[108] |
|
LAMP2 |
P13473 |
Decreased |
ELISA |
[108] |
|
LAMP1 |
P11279 |
Decreased |
Immunoblotting |
[109] |
aThe protein symbol indicates the human protein designated by UniProtKB. bAbbreviations: LC-MS/MS, liquid chromatography-tandem mass spectrometry; MRM-LC-MS/MS, multiple reaction monitoring-liquid chromatography-tandem mass spectrometry; HPLC-MS/MS, high-performance liquid chromatography-tandem mass spectrometry; ELISA, enzyme-linked immunosorbent assay; LC-IMS-MS, liquid chromatography-ion mobility spectrometry-mass spectrometry; LC-MRM, liquid chromatography-multiple reaction monitoring; LC-PRM-MS, liquid chromatography-parallel reaction monitoring-mass spectrometry; SIMOA, single molecule array.
5. In lines 190-191, it is indicated that “synthesis of ATP”, “function of mitochondria” and “release of neurotransmitter” were inhibited (Figure 2B). In Figure 2(B), blue indicates inhibition, but the color intensity of these three is not consistent. Does the color intensity have a special meaning, such as whether it represents the degree of inhibition? If so, should an explanation be made.
à As you commented, the color intensity has a meaning in prediction. The color represents activation (orange) or inhibition (blue), and the color intensity reflects the confidence of the prediction rather the degree of activation or inhibition. This confidence is influenced by the number of factors contributing to the inhibitory effect. For instance, the inhibition of “synthesis of ATP” is supported by four inhibitory factors among six total linked factors. In contrast, the inhibition of “function of mitochondria” involves fewer inhibitory factors, resulting in a lighter blue. In this analysis, we focused on the prediction outcomes themselves, demonstrating that alterations in proteins in the CSF of patients with PD closely reflect the pathological conditions of PD. To avoid misunderstanding, we edited each figure legend to include an explanation of color intensity.
Line 174, page 6 | Line 225, page 10 | Line 268, page 12
The color intensity reflects the confidence of the prediction.
6. In Introduction section, the authors described “Parkinson’s disease (PD) is a progressive neurodegenerative disorder...”, it is recommended to add a paragraph to describe the unique advantages of cerebrospinal fluid (CSF) compared to other bodily fluids, thus strengthening the rationale for its selection.
à In line 57-62, we briefly discussed the advantages of analyzing CSF in PD research. However, according to your suggestion, we have expanded on this section to further emphasize the importance of CSF analysis in PD research, particularly in lines 58 and 65. We have also dedicated a paragraph for the content of advantages of CSF compared to other bodily fluids.
Line 58-61, page 2
Unlike peripheral fluids, CSF directly interfaces with the extracellular space of the brain, allowing an unrestricted two-way exchange of molecules between these regions [7]. In contrast, brain-derived proteins are typically not detected in blood-derived samples [24].
Line 65-66, page 2
In addition, CSF from PD patients was found to include toxic factors for dopaminergic neurons [26], highlighting its usefulness for PD research.
Additional References
- Kwon, E.H.; Tennagels, S. Update on CSF biomarkers in Parkinson’s disease. Biomolecules. 2022, 12, 329.
- Constantinescu, R.; Mondello, S. Cerebrospinal fluid biomarker candidates for parkinsonian disorders. Front Neurol. 2013, 3, 187.
- Le, W.-D.; Rowe, D.B. Effects of cerebrospinal fluid from patients with Parkinson disease on dopaminergic cells. Arch Neurol. 1999, 56, 194-200.
7. In section “3. CSF miRNA in PD”, various PD-related miRNAs are mentioned, but the specific criteria and rationale for selecting these miRNAs are not provided. It would be beneficial to elaborate on the exact standards used to select these miRNAs and to discuss in depth how these miRNAs are involved in the biological processes of PD.
à The miRNAome dataset was collected based on specific criteria (p < 0.05 and the alteration ratio exceeding 20% compared to the control). With this dataset, IPA software enables the construction of interactive molecular networks based on existing literature, as illustrated in Figure 1B. Here, we focused on four key miRNAs, miR-16-5p, miR-7a-5p, miR-15 and miR-17-5p that demonstrated either inhibitory or activating effects on biological functions. Detailed explanations of these four key miRNAs have been provided in the manuscript, addressing your comments.
Line 156-164, page 5
These findings are supported by previous studies. Specifically, miR-7a-5p, which is associated with ATP synthesis, has been reported to suppress cell proliferation upon upregulation, whereas its downregulation reduced apoptosis in non-small-cell lung cancer [91]. Similarly, miR-16-5p was shown to enhance mitochondrial function when its expression level was reduced in bladder cancer [92]. Furthermore, overexpression of mir-15 decreased ATP level in rat ventricular myocyte [93]. Finally, miR-17-5p overexpression impaired TGF-beta signaling, leading to neurodegeneration in SH-SY5Y cells [94]. Taken together, these altered levels in CSF are indicative of biological processes underlying PD.
Additional References
- Li, Q.; Wu, X. MicroRNA‐7‐5p induces cell growth inhibition, cell cycle arrest and apoptosis by targeting PAK2 in non‐small cell lung cancer. FEBS open bio. 2019, 9, 1983-1993.
- Li, H.-J.; Sun, X.-M. LncRNA UCA1 promotes mitochondrial function of bladder cancer via the MiR-195/ARL2 signaling pathway. Cell Physiol Biochem. 2017, 43, 2548-2561.
- Nishi, H.; Ono, K. MicroRNA-15b Modulates Cellular ATP Levels and Degenerates Mitochondria via Arl2 in Neonatal Rat Cardiac Myocytes 2. J Biol Chem. 2010, 285, 4920-4930.
- Wang, H.; Liu, J. miR-106b aberrantly expressed in a double transgenic mouse model for Alzheimer's disease targets TGF-β type II receptor. Brain Res. 2010, 1357, 166-174.

Reviewer 2 Report
Comments and Suggestions for Authors
This is a review as well as authors’ own analysis combining published miRNAomic and proteomic data from PD using a computational method. The paper is well written and very current. My criticism is minor.
The original data on which the analysis is based are often limited to single reports which were not yet corroborated by others. This should be mentioned as a limitation. Besides, since this is a review, some criticism of published studies as well as pointing out discrepancies, when present, should be included.
Because this is a review on miRNAs in PD a short paragraph in the Introduction on miRNAs seems warranted (such information on PD is provided). Granted, there is information in the body of the paper, but it should be moved into Introduction.
Author Response
Thank you for insightful comments and suggestions. According to your comments, we carefully checked and revised the manuscript. The responses were provided right below each comment, with your comment in black, our responses in blue, and the changes made in the manuscript were colored in red.
Comments for Suggestions for Authors:
This is a review as well as authors’ own analysis combining published miRNAomic and proteomic data from PD using a computational method. The paper is well written and very current. My criticism is minor
.
1. The original data on which the analysis is based are often limited to single reports which were not yet corroborated by others. This should be mentioned as a limitation. Besides, since this is a review, some criticism of published studies as well as pointing out discrepancies, when present, should be included.
à As you commented, our data collection is mostly based on individual reports. Therefore, we addressed this limitation and discussed the molecules with discrepancies in the manuscript.
Line 288-294, page 11
While this review highlights the potential of integrating multi-omics for PD analysis, it also has limitations. Data collection primarily relied on individual studies that were not corroborated by other research, emphasizing the need for follow-up studies. For instance, discrepancies in findings related to molecules such as miR-127-3p [55,89], and miR-136-3p [86,87], miR-433 [55,86] have hindered efforts to clarify mechanisms underlying PD. These inconsistencies suggest that more robust and reproducible studies are essential to resolve conflicting results and improve our understanding of molecular targets in PD.
Additional references
- Burgos, K.; Malenica, I. Profiles of extracellular miRNA in cerebrospinal fluid and serum from patients with Alzheimer's and Parkinson's diseases correlate with disease status and features of pathology. PLoS One. 2014, 9, e94839.
- Gui, Y.; Liu, H. Altered microRNA profiles in cerebrospinal fluid exosome in Parkinson disease and Alzheimer disease. Oncotarget. 2015, 6, 37043-37053.
- Tong, G.; Zhang, P. Diagnostic test to Identify Parkinson’s disease from the blood sera of Chinese population: A cross‐sectional study. Parkinsons Dis. 2022, 2022, 8683877.
- Caldi Gomes, L.; Roser, A.E. MicroRNAs from extracellular vesicles as a signature for Parkinson's disease. Clin Transl Med. 2021, 11, e357.
2. Because this is a review on miRNAs in PD a short paragraph in the Introduction on miRNAs seems warranted (such information on PD is provided). Granted, there is information in the body of the paper, but it should be moved into Introduction.
à According to your comment, we moved the content regarding miRNAs from the body of the paper into the Introduction section. This part is in line 70-75, page 2 and is highlighted in red.
Line 70-75, page 2
Among the biomolecules of interest, miRNAs have emerged as important regulators of gene expression and translation [31]. These small non-coding RNAs, approximately 22 nucleotides in length [32], were first identified in Caenorhabditis elegans and regulate gene expression by interacting with complementary sequences on target mRNA through anti-sense RNA-RNA interactions [33]. miRNAs are implicated in various biological processes from the development, differentiation, proliferation, cell death, and immune systems [34].

Round 2
Reviewer 1 Report
Comments and Suggestions for Authors
The authors have well replied my comments, I recommend accept.